# Clinicopathologic features and outcomes of bilateral lacrimal gland lesions

Lvfu He[1,2], Weimin He[1]*

1 Department of Ophthalmology, West China Hospital of Sichuan University, Chengdu, Sichuan Province, China, 2 Department of Ophthalmology, The Third Hospital of Mianyang, Mianyang, Sichuan Province, China

* hewm888@hotmail.com

**Data Availability Statement:** All relevant data are within the manuscript and its Supporting Information files.

**Funding:** The author(s) received no specific funding for this work.

## Abstract

### Background

The present study reviewed the clinicopathological features and outcomes of bilateral lacrimal gland lesions.

### Methods

The data of 113 patients who underwent lacrimal gland biopsy at the West China Hospital of Sichuan University, China, between January 1, 2010, and December 31, 2021, are presented in this case series. The patients all presented with bilateral lacrimal gland lesions. The collected data included patient demographics, clinical features, the results of laboratory examinations, imaging presentations, histopathological diagnoses, treatments, and outcomes.

### Results

The mean age of the 113 enrolled patients was 47.4 ± 14.9 years (range, 11–77 years) with a predominance of females (54.9%, n = 62). The lacrimal gland was the source of the majority of biopsy tissue (98.2%, n = 111). The most prevalent etiology was immunoglobulin G4-related ophthalmic disease (IgG4-ROD) (32.7%, n = 37), followed by idiopathic orbital inflammation (IOI) (28.3%, n = 32), mucosa-associated lymphoid tissue (MALT) lymphoma (17.7%, n = 20), reactive lymphoid hyperplasia (RLH) (10.6%, n = 12), and mantle cell lymphoma (4.4%, n = 5). Patients with IOI were significantly younger than those with IgG4-ROD and MALT lymphoma (t = 2.932, $P$ = 0.005; t = 3.865, $P$<0.001, respectively). Systemic symptoms were more prevalent among patients with IgG4-ROD ($\chi2$ = 7.916, $P$ = 0.005). The majority of patients were treated with surgery (53.1%, n = 60), with surgery combined with corticosteroid therapy (21.2%, n = 24) being the second most common treatment. The majority of patients (91.2%, n = 103) attained complete resolution, stable disease, or significant improvement.

### Conclusion

In conclusion, there are several aetiologies associated with bilateral lacrimal gland lesions, the most prevalent being IgG4-ROD, IOI, and MALT lymphoma. Systemic symptoms were

**Competing interests:** The authors have declared that no competing interests exist.

more common in patients with IgG4-ROD. The majority of patients who presented with bilateral lesions of the lacrimal glands responded satisfactorily to treatment, with favorable results.

## Introduction

Most lacrimal gland lesions are unilateral and only a minority of patients present with bilateral disease [1]. These symptoms may be specific to the lacrimal gland or could indicate a systemic condition that is affecting the orbit. Multiple studies have provided detailed descriptions of the clinical characteristics and results of bilateral lacrimal gland lesions [1, 2]. For instance, Huang et al. found that immunoglobulin G4-related ophthalmic disease (IgG4-ROD), idiopathic orbital inflammatory disease (IOID), and lymphoma were the most common causes of the lesions [2]. Furthermore, thyroid eye disease (TED) can affect both lacrimal glands symmetrically. However, surgical intervention is usually not pursued due to the availability of a definitive diagnosis. Currently, there have been no assessments of the clinicopathological features of bilateral lacrimal gland lesions that have been biopsied in Asian populations. Therefore, the objective of this study was to examine the clinicopathological features as well as outcomes of patients with bilateral lacrimal gland lesions who received biopsies at the West China Hospital of Sichuan University, located in China.

## Materials and methods

### Patients

This study was conducted as a case series. Included in this study were patients with bilateral lacrimal gland lesions who had lacrimal gland biopsy at our hospital from January 1, 2010, to December 31, 2021. The diagnosis of bilateral lacrimal gland lesions was established by combining clinical symptoms, laboratory investigations, imaging findings, and biopsy analysis. Patients who underwent diagnostic lacrimal gland biopsy and those who had previously received medical or surgical intervention for the lacrimal gland were not included in the study.

### Data collection

Data regarding patient demographics, clinical characteristics, serological and radiographic results, histological diagnosis, therapies, and prognoses were collected. For patients who experienced a relapse, only their initial occurrence was considered in the analysis. The follow-up period extended from the date of pathological diagnosis until December 31, 2022.

### Ethical approval

The study was carried out in compliance with the ethical principles outlined in the Declaration of Helsinki and received approval from the Review Board of The West China Hospital of Sichuan University. All patients or their relatives provided informed verbal agreement for their participation in this study, and their confidentiality was preserved.

### Statistical analysis

All data were analyzed using SPSS software version 27.0 (IBM Corp., Armonk, N.Y, USA). Continuous variables are shown as mean ± standard deviation and were analyzed using Student's t-test or the Mann-Whitney U test. Categorical variables were analyzed using chi-

squared or Fisher's exact tests and presented as frequencies and percentages. The recurrence rates were assessed using the Kaplan-Meier method and compared between groups using the log-rank test. Statistically significant values were defined as $P < 0.05$.

## Results

### Demographic characteristics

A total of 113 patients were evaluated in this study. The average age was $47.4 \pm 14.9$ years (ranging from 11 to 77 years), with 51 patients (45.1%) being male. The predominant ethnicity, including 95.5% (n = 108), was Han. Furthermore, the average length of time that symptoms were present was $35.9 \pm 48.5$ months (ranging from 0.5 to 240 months) (Table 1).

### Clinical presentations

Table 2 shows the patient's diagnosis of bilateral lacrimal gland lesions along with the accompanying ocular symptoms. Lid swelling (72.6%, n = 82), the presence of a palpable mass (61.9%, n = 70), decreased ocular movements (DOM) (23.9%, n = 27), conjunctival injection (18.6%, n = 21), proptosis (17.7%, n = 20), and lid hyperemia (14.2%, n = 16) were the most prevalent ophthalmic signs and symptoms. Less common symptoms were globe dystopia (n = 3), keratopathy (n = 2), lagophthalmos (n = 2), and dry eye (n = 1). Lid swelling was found to be substantially correlated with IgG4-ROD ($\chi2 = 7.636$, $P = 0.006$), whereas palpable masses were most frequently observed in cases of mucosa-associated lymphoid tissue (MALT) lymphoma ($\chi2 = 11.263$, $P < 0.001$).

A total of 49 individuals (43.4%) developed extra systemic symptoms. Patients with IgG4-ROD had a higher frequency of systemic symptoms ($\chi2 = 7.916$, $P = 0.005$). Table 3 summarises the systemic symptoms and diagnosis related to bilateral lacrimal gland lesions.

### Laboratory investigations

Every patient received a laboratory investigation. Out of the 31 individuals who underwent blood IgG4 testing, 17 (54.8%) showed increased levels of IgG4. All of these patients, except for one with idiopathic orbital inflammation (IOI), were diagnosed with IgG4-ROD. The levels of anti-RNP/Sm antibodies, anti-SSA antibodies, and IgM were elevated in one patient with Sjogren's syndrome. A patient diagnosed with MALT lymphoma displayed increased levels of

**Table 1. Demographic characteristics and disease features.**

| | All of the patients |
|---|---|
| Number | 113 |
| Age, mean years (range) | $47.4 \pm 14.9(11-77)$ |
| Sex | |
| Male, number (%) | 51(45.1%) |
| Female, number (%) | 62(54.9%) |
| Race | |
| Han, number (%) | 108(95.5%) |
| Yi, number (%) | 2(1.8%) |
| Zang, number (%) | 2(1.8%) |
| Qiang, number (%) | 1(0.9%) |
| Symptom presenting period, mean months (range) | $35.9 \pm 48.5(0.5-240)$ |
| Systemic symptoms, number (%) | 49(43.4%) |
| Allergy history, number (%) | 7(6.2%) |

**Table 2. Ophthalmic signs and symptoms of patients with bilateral lacrimal gland lesions by diagnosis.**

| Diagnosis | Lid swelling | Palpable mass | DOM | Conjunctival injection | Proptosis | Lid hyperemia | Periorbital swelling | Periorbital pain | Visual loss | Lacrimation | Chemosis | Ptosis |
|---|---|---|---|---|---|---|---|---|---|---|---|---|
| Total(n = 113) | 82(72.6) | 70(61.9) | 27(23.9) | 21(18.6) | 20(17.7) | 16(14.2) | 13(11.5) | 13(11.5) | 15(13.3) | 15(13.3) | 11(9.7) | 6(5.3) |
| Lacrimal gland origin(n = 111) | 80(72.1) | 69(62.2) | 25(22.5) | 19(17.1) | 18(16.2) | 15(13.5) | 12(10.8) | 12(10.8) | 13(11.7) | 15(13.5) | 10(9.0) | 6(5.4) |
| Inflammations(n = 36) | 24(66.7) | 16(44.4) | 5(13.9) | 5(13.9) | 2(5.6) | 7(19.4) | 4(11.1) | 4(11.1) | 6(16.7) | 1(2.8) | 1(2.8) | 0(0.0) |
| IOI(n = 32) | 20(62.5) | 14(43.8) | 4(12.5) | 4(12.5) | 2(6.3) | 6(18.8) | 4(12.5) | 3(9.4) | 5(15.6) | 1(3.1) | 1(3.1) | 0(0.0) |
| Sjogren's(n = 2) | 2(100) | 1(50.0) | 0(0.0) | 0(0.0) | 0(0.0) | 0(0.0) | 0(0.0) | 0(0.0) | 0(0.0) | 0(0.0) | 0(0.0) | 0(0.0) |
| Amyloidosis(n = 1) | 1(100) | 0(0.0) | 1(100) | 1(100) | 0(0.0) | 0(0.0) | 0(0.0) | 0(0.0) | 1(100) | 0(0.0) | 0(0.0) | 0(0.0) |
| Still's disease(n = 1) | 1(100) | 1(100) | 0(0.0) | 0(0.0) | 0(0.0) | 1(100) | 0(0.0) | 1(100) | 0(0.0) | 0(0.0) | 0(0.0) | 0(0.0) |
| IgG4-ROD(n = 37) | 33(89.2) | 23(62.2) | 13(35.1) | 9(24.3) | 10(27.0) | 4(10.8) | 4(10.8) | 4(10.8) | 3(8.1) | 7((18.9) | 7(18.9) | 3((8.1) |
| MALT lymphoma(n = 20) | 11(55.0) | 19(95.0) | 7(35.0) | 3(15.0) | 5(25.0) | 1(5.0) | 4(20.0) | 4(20.0) | 4(20.0) | 3(15.0) | 2(10.0) | 3(15.0) |
| Mantle cell lymphoma(n = 5) | 2(40.0) | 4(80.0) | 0(0.0) | 2(40.0) | 0(0.0) | 1(20.0) | 0(0.0) | 0(0.0) | 0(0.0) | 1(20.0) | 0(0.0) | 0(0.0) |
| RLH(n = 12) | 9(75.0) | 6(50.0) | 0(0.0) | 0(0.0) | 1(8.3) | 2(16.7) | 0(0.0) | 0(0.0) | 0(0.0) | 3(25.0) | 0(0.0) | 0(0.0) |
| Structural(n = 1) | 1(100) | 1(100) | 0(0.0) | 0(0.0) | 0(0.0) | 0(0.0) | 0(0.0) | 0(0.0) | 0(0.0) | 0(0.0) | 0(0.0) | 0(0.0) |
| Dacryops(n = 1) | 1(100) | 1(100) | 0(0.0) | 0(0.0) | 0(0.0) | 0(0.0) | 0(0.0) | 0(0.0) | 0(0.0) | 0(0.0) | 0(0.0) | 0(0.0) |
| NON-lacrimal gland origin (n = 2) | 2(100) | 1(50.0) | 2(100) | 2(100) | 2(100) | 1(50.0) | 1(50.0) | 1(50.0) | 2(100) | 0(0.0) | 1(50.0) | 0(0.0) |
| Inflammations(n = 1) | 1(100) | 1(100) | 1(100) | 1(100) | 1(100) | 1(100) | 0(0.0) | 0(0.0) | 1(100) | 0(0.0) | 0(0.0) | 0(0.0) |
| Xanthogranulomatous disease(n = 1) | 1(100) | 1(100) | 1(100) | 1(100) | 1(100) | 1(100) | 0(0.0) | 0(0.0) | 1(100) | 0(0.0) | 0(0.0) | 0(0.0) |
| Neoplastic(n = 1) | 1(100) | 0(0.0) | 1(100) | 1(100) | 1(100) | 0(0.0) | 1(100) | 1(100) | 1(100) | 0(0.0) | 1(100) | 0(0.0) |
| Erdheim Chester disease (n = 1) | 1(100) | 0(0.0) | 1(100) | 1(100) | 1(100) | 0(0.0) | 1(100) | 1(100) | 1(100) | 0(0.0) | 1(100) | 0(0.0) |

All results expressed as: n (%).

IOI Idiopathic orbital inflammation, IgG4-ROD Immunoglobulin G4-related ophthalmic disease, MALT Mucosa associated lymphoid tissue, RLH Reactive lymphoid hyperplasia, DOM Decreased ocular movements.

**Table 3. Systemic symptoms of patients with bilateral lacrimal gland lesions by diagnosis.**

| Diagnosis | Total number of patients with condition | Number of patients with systemic symptoms | Respiratory symptoms | Cardio-vascular symptoms | Endocrine symptoms | Digestive symptoms | Urinary symptoms | Lympha-denopathy | Connective tissue symptoms | Swollen salivary glands | Sinusitis | Rash |
|---|---|---|---|---|---|---|---|---|---|---|---|---|
| IOI | 32 | 6(18.8) | 1(3.1) | 1(3.1) | 2(6.3) | 0(0.0) | 1(3.1) | 1(3.1) | 1(3.1) | 1(3.1) | 0(0.0) | 0(0.0) |
| Sjogren's | 2 | 1(50.0) | 0(0.0) | 1(50.0) | 0(0.0) | 0(0.0) | 0(0.0) | 0(0.0) | 1(50.0) | 0(0.0) | 0(0.0) | 0(0.0) |
| Xanthogranulomatous disease | 1 | 1(100) | 0(0.0) | 1(100) | 0(0.0) | 0(0.0) | 1(100) | 0(0.0) | 0(0.0) | 0(0.0) | 0(0.0) | 0(0.0) |
| Still's disease | 1 | 1(100) | 0(0.0) | 0(0.0) | 0(0.0) | 0(0.0) | 0(0.0) | 1(100) | 1(100) | 0(0.0) | 0(0.0) | 1(100) |
| Amyloidosis | 1 | 0(0.0) | 0(0.0) | 0(0.0) | 0(0.0) | 0(0.0) | 0(0.0) | 0(0.0) | 0(0.0) | 0(0.0) | 0(0.0) | 0(0.0) |
| IgG4-ROD | 37 | 23(62.2) | 7(18.9) | 4(10.8) | 6(16.2) | 3(8.1) | 2(5.4) | 3(8.1) | 1(2.7) | 6(16.2) | 3(8.1) | 1(2.7) |
| MALT lymphoma | 20 | 8(40.0) | 2(10.0) | 2(10.0) | 1(5.0) | 3(15.0) | 0(0.0) | 0(0.0) | 1(5.0) | 1(5.0) | 0(0.0) | 1(5.0) |
| Mantle cell lymphoma | 5 | 3(60.0) | 0(0.0) | 0(0.0) | 0(0.0) | 0(0.0) | 0(0.0) | 2(40.0) | 0(0.0) | 3(60.0) | 0(0.0) | 0(0.0) |
| RLH | 12 | 4(33.3) | 1(8.3) | 2(16.7) | 0(0.0) | 1(8.3) | 3(25.0) | 0(0.0) | 0(0.0) | 0(0.0) | 0(0.0) | 0(0.0) |
| Dacryops | 1 | 1(100) | 0(0.0) | 1(100) | 0(0.0) | 0(0.0) | 0(0.0) | 0(0.0) | 0(0.0) | 0(0.0) | 0(0.0) | 0(0.0) |
| Erdheim-Chester disease | 1 | 1(100) | 1(100) | 1(100) | 0(0.0) | 0(0.0) | 1(100) | 0(0.0) | 0(0.0) | 0(0.0) | 0(0.0) | 0(0.0) |
| Total | 113 | 49(43.4) | 12(10.6) | 13(11.5) | 9(8.0) | 7(6.2) | 8(7.1) | 7(6.2) | 5(4.4) | 11(9.7) | 3(2.7) | 3(2.7) |

All results expressed as: n (%).

IOI Idiopathic orbital inflammation, IgG4-ROD Immunoglobulin G4-related ophthalmic disease, MALT Mucosa associated lymphoid tissue, RLH Reactive lymphoid hyperplasia.

C3, C4, rheumatoid factor, anti-SSA/SSB antibodies, anti-Scl-70 antibodies, and anti-Ro-52 antibodies. On the other hand, a patient identified with Still's disease had elevated levels of C3, anti-SSA antibodies, and anti-Ro-52 antibodies.

## Radiological investigations

Orbital imaging was performed in 107 patients (94.7%). Of these, 63 patients (55.8%) underwent computed tomography (CT) scans, 36 patients (31.9%) underwent magnetic resonance imaging (MRI), and 8 patients (7.1%) had both performed. Out of the total number of patients with IOI, 6 individuals (5.3%) did not receive any imaging evaluation. The majority of lacrimal gland lesions exhibited masses of moderate density on CT scans. Most patients with lacrimal gland lesions displayed moderate signal intensities on both T1 and T2-weighted imaging and showed elevated signals with contrast-enhanced scans on MRI. Furthermore, 2 patients underwent chest CT scans, which led to the diagnosis of Still's disease (n = 1) and mantle cell lymphoma (n = 1), respectively. A PET/CT scan was performed on a single patient with IgG4-ROD.

Enlargement of the bilateral lacrimal glands was seen on imaging scans of 105 patients (92.9%). Moreover, 21 patients (18.6%) had involvement of other orbital structures, including parotid glands, submandibular glands, optic nerve, infraorbital nerve, supraorbital fissure, extraocular muscles, and lymph nodes. Of these, 13 patients had been diagnosed with IgG4-ROD, 3 with IOI, 3 with MALT lymphoma, 1 with RLH, and 1 with mantle cell lymphoma.

## Histopathological classification

The histopathological classification of the bilateral lacrimal gland lesions is shown in Table 4. 32.7% (n = 37) of the patient population showed IgG4-ROD, and 28.3% (n = 32) developed IOI. The remaining prevalent specific aetiologies were MALT lymphoma (17.7%, n = 20), RLH (10.6%, n = 12), and mantle cell lymphoma (4.4%, n = 5). Overall, patients with IOI were significantly younger (mean, 38.9 years) than those with IgG4-ROD or MALT lymphoma (t = 2.932, $P$ = 0.005; t = 3.865, $P$<0.001, respectively). Comparing patients with other diseases, lymphoma patients tended to be older (mean, 55.8 years). The duration of symptoms in IOI, IgG4-ROD, MALT lymphoma, and RLH was not significantly different (all $P$>0.05).

Except for IgG4-ROD (35.1%, 13 of 37), dacryops (0%, 0 of 1), xanthogranulomatous disease (0%, 0 of 1), and Erdheim-Chester disease (0%, 0 of 1), a female predominance was observed for most of the conditions.

## Treatment, response, and outcomes

Table 5 presents information on the treatment and outcomes of bilateral lacrimal gland lesions in terms of diagnosis. Data on treatment results were obtained for 108 patients (95.6%), however, five patients' data (4.4%) were lost. The cohort had a mean follow-up length of 72.4 ± 40.7 months (range, 12–149 months). Sixty-eight patients (60.2%) underwent bilateral biopsy, with an average time interval between the bilateral surgeries of 155.1 ± 448.3 days (0.0–2974.0 days), while 45 patients (39.8%) underwent unilateral biopsy (Table 4). Combination therapy was used to treat 53 patients (46.9%), with surgery and corticosteroids being the most common treatment (21.2%, n = 24). Furthermore, surgery was utilized in conjunction with chemotherapy, radiation, or immunosuppressive. Prednisone acetate, vindesine, azathioprine, methotrexate, cyclosporin, ibutinib, cyclophosphamide, hydroxychloroquine, daunorubicin, and rituximab were among the drugs utilized.

**Table 4. Distribution of pathology and demographic data for bilateral lacrimal gland lesions.**

| Diagnosis | NO.(%) | Male(%) | Mean age (Range) | Bilateral biopsy(%) | Recurrence (%) |
|---|---|---|---|---|---|
| Total | 113(100) | 51(45.1) | 47.4 (11–77) | 68(60.2) | 9(8.0) |
| **Lacrimal gland origin** | 111(98.2) | 49(44.1) | 47.4(11–77) | 66(59.5) | 9(8.1) |
| **Inflammations** | 36(31.9) | 8(22.2) | 39.5(16–76) | 19(52.8) | 2(5.6) |
| IOI | 32(28.3) | 8(25.0) | 38.9(16–76) | 18(56.3) | 2(6.3) |
| Sjogren's | 2(1.8) | 0(0.0) | 38.5(25–52) | 0(0.0) | 0(0.0) |
| Amyloidosis | 1(0.9) | 0(0.0) | 50.0 | 1(100) | 0(0.0) |
| Still's disease | 1(0.9) | 0(0.0) | 50.0 | 0(0.0) | 0(0.0) |
| **IgG4-ROD** | 37(32.7) | 24(64.9) | 48.6(15–74) | 22(59.5) | 4(10.8) |
| **MALT lymphoma** | 20(17.7) | 9(45.0) | 53.4(32–77) | 12(60.0) | 3(15.0) |
| **RLH** | 12(10.6) | 5(41.7) | 48.1(11–75) | 9(75.0) | 0(0.0) |
| **Mantle cell lymphoma** | 5(4.4) | 2(40.0) | 65.4(51–75) | 3(60.0) | 0(0.0) |
| **Structural** | 1(0.9) | 1(100) | 69.0 | 1(100) | 0(0.0) |
| Dacryops | 1(0.9) | 1(100) | 69.0 | 1(100) | 0(0.0) |
| **NON-lacrimal gland origin** | 2(1.8) | 2(100) | 48.0(39–57) | 2(100) | 0(0.0) |
| **Inflammations** | 1(0.9) | 1(100) | 39.0 | 1(100) | 0(0.0) |
| Xanthogranulomatous disease | 1(0.9) | 1(100) | 39.0 | 1(100) | 0(0.0) |
| **Neoplastic** | 1(0.9) | 1(100) | 57.0 | 1(100) | 0(0.0) |
| Erdheim Chester disease | 1(0.9) | 1(100) | 57.0 | 1(100) | 0(0.0) |

All results expressed as: n (%).

IOI Idiopathic orbital inflammation, IgG4-ROD Immunoglobulin G4-related ophthalmic disease, MALT Mucosa associated lymphoid tissue, RLH Reactive lymphoid hyperplasia.

Recurrence of bilateral lacrimal gland lesions occurred in 9 patients (8.0%) and three patients (2.7%) died. The range of the recurrence-free duration was 14–108 months, with an average of 42.0 ± 30.3 months. Log-rank testing revealed that individuals with IOI experienced recurrence earlier than those with IgG4-ROD (Fig 1A) and MALT lymphoma (Fig 1B). However, there were no significant differences between IgG4-ROD and MALT lymphoma ($P = 0.123$). MALT lymphoma was the cause of death for each of the deceased individuals, who also suffered from organ failure. 23.0% of patients (n = 26) showed significant improvement, 37 patients (32.7%) showed complete resolution, and 40 patients (35.4%) had stable disease. Disease progression was noted in five patients (4.4%), including 3 with IgG4-ROD, 1 with MALT lymphoma, and 1 with mantle cell lymphoma.

## Discussion

Lacrimal gland lesions can occur on both sides of the body and may be either confined to a specific area or linked to systemic disorders. In addition, the lesions show a wide range of causes. In certain cases, the cause of the condition can be determined without the need for a biopsy. This can be done by evaluating the symptoms of the individual, examining imaging results, and analyzing laboratory tests. An example of this is individuals with TED. Some lacrimal gland lesions are initially managed with hormone therapy in clinical practice. If hormone therapy fails to produce the desired results, surgical intervention is then considered. Surgery not only eliminates the lesion but also enables the precise determination of its cause through pathological investigation. Clarifying the treatment direction is beneficial to prevent potential hazards such as misdiagnosis, improper therapy, and adverse consequences linked to the prolonged use of glucocorticoids.

**Table 5. Treatment and outcomes of patients with bilateral lacrimal gland lesions by diagnosis.**

| Diagnosis | Treatment | | | | | | Outcomes | | | | |
|---|---|---|---|---|---|---|---|---|---|---|---|
| | Surgical | Combined surgical and corticosteroids | Combined surgical and corticosteroids and immunosuppressant | Combined surgical and chemotherapy (or rituximab) | Combined surgical and radiation therapy | Combined surgical and chemotherapy and rituximab and immunosuppressant | Resolved | Stable | Improved | Progressed | Unknown |
| Total(n = 113) | 60(53.1) | 24(21.2) | 16(14.2) | 8(7.1) | 4(3.5) | 2(1.8) | 37(32.7) | 40(35.4) | 26(23.0) | 5(4.4) | 5(4.4) |
| Lacrimal gland origin (n = 111) | 58(52.3) | 24(21.6) | 16(14.4) | 8(7.2) | 4(3.6) | 2(1.8) | 36(32.4) | 40(36.0) | 25(22.5) | 5(4.5) | 5(4.5) |
| **Inflammations** (n = 36) | 33(91.7) | 1(2.8) | 2(5.6) | 0(0.0) | 0(0.0) | 0(0.0) | 21(58.3) | 8(22.2) | 5(13.9) | 0(0.0) | 2(5.6) |
| IOI(n = 32) | 32(100) | 0(0.0) | 0(0.0) | 0(0.0) | 0(0.0) | 0(0.0) | 20(62.5) | 8(25.0) | 2(6.3) | 0(0.0) | 2(6.3) |
| Sjögren's(n = 2) | 0(0.0) | 1(50.0) | 1(50.0) | 0(0.0) | 0(0.0) | 0(0.0) | 0(0.0) | 0(0.0) | 2(100) | 0(0.0) | 0(0.0) |
| Amyloidosis (n = 1) | 1(100) | 0(0.0) | 0(a0.0) | 0(0.0) | 0(0.0) | 0(0.0) | 1(100) | 0(0.0) | 0(0.0) | 0(0.0) | 0(0.0) |
| Still's disease (n = 1) | 0(0.0) | 0(0.0) | 1(100) | 0(0.0) | 0(0.0) | 0(0.0) | 0(0.0) | 0(0.0) | 1(100) | 0(0.0) | 0(0.0) |
| **IgG4-ROD**(n = 37) | 0(0.0) | 23(62.2) | 14(37.8) | 0(0.0) | 0(0.0) | 0(0.0) | 4(10.8) | 18(48.6) | 12(32.4) | 3(8.1) | 0(0.0) |
| **MALT lymphoma** (n = 20) | 9(45.0) | 0(0.0) | 0(0.0) | 8(40.0) | 4(20.0) | 0(0.0) | 5(25.0) | 7(35.0) | 7(35.0) | 1(5.0) | 0(0.0) |
| **Mantle cell lymphoma**(n = 5) | 3(60.0) | 0(0.0) | 0(0.0) | 0(0.0) | 0(0.0) | 2(40.0) | 0(0.0) | 2(40.0) | 1(20.0) | 1(20.0) | 1(20.0) |
| RLH(n = 12) | 12(100) | 0(0.0) | 0(0.0) | 0(0.0) | 0(0.0) | 0(0.0) | 5(41.7) | 5(41.7) | 0(0.0) | 0(0.0) | 2(16.7) |
| **Structural**(n = 1) | 1(100) | 0(0.0) | 0(0.0) | 0(0.0) | 0(0.0) | 0(0.0) | 1(100) | 0(0.0) | 0(0.0) | 0(0.0) | 0(0.0) |
| Dacryops(n = 1) | 1(100) | 0(0.0) | 0(0.0) | 0(0.0) | 0(0.0) | 0(0.0) | 1(100) | 0(0.0) | 0(0.0) | 0(0.0) | 0(0.0) |
| **NON-lacrimal gland origin**(n = 2) | 2(100) | 0(0.0) | 0(0.0) | 0(0.0) | 0(0.0) | 0(0.0) | 1(50.0) | 0(0.0) | 1(50.0) | 0(0.0) | 0(0.0) |
| **Inflammations** (n = 1) | 1(100) | 0(0.0) | 0(0.0) | 0(0.0) | 0(0.0) | 0(0.0) | 1(100) | 0(0.0) | 0(0.0) | 0(0.0) | 0(0.0) |
| Xanthogranulomatous disease(n = 1) | 1(100) | 0(0.0) | 0(0.0) | 0(0.0) | 0(0.0) | 0(0.0) | 1(100) | 0(0.0) | 0(0.0) | 0(0.0) | 0(0.0) |
| **Neoplastic**(n = 1) | 1(100) | 0(0.0) | 0(0.0) | 0(0.0) | 0(0.0) | 0(0.0) | 0(0.0) | 0(0.0) | 1(100) | 0(0.0) | 0(0.0) |
| Erdheim Chester disease(n = 1) | 1(100) | 0(0.0) | 0(0.0) | 0(0.0) | 0(0.0) | 0(0.0) | 0(0.0) | 0(0.0) | 1(100) | 0(0.0) | 0(0.0) |

All results expressed as: n (%).

IOI Idiopathic orbital inflammation, IgG4-ROD Immunoglobulin G4-related ophthalmic disease, MALT Mucosa associated lymphoid tissue, RLH Reactive lymphoid hyperplasia.

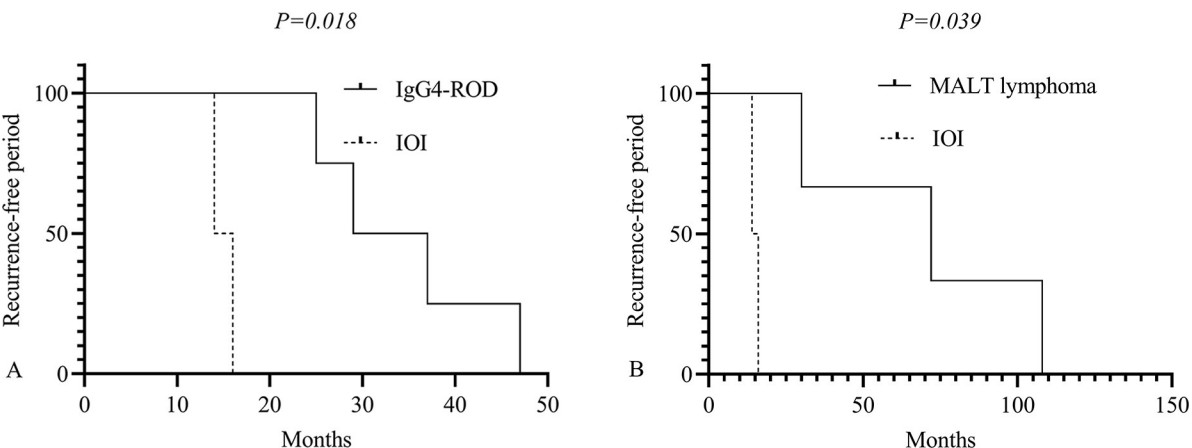

**Fig 1.** Recurrence-free period of patients with IgG4-ROD and IOI (A), MALT lymphoma and IOI (B). IOI Idiopathic orbital inflammation, IgG4-ROD immunoglobulin G4-related ophthalmic disease, IMALT Mucosa associated lymphoid tissue.

The present study identified several clinical characteristics that were commonly associated with particular diseases. Additionally, it was discovered that IgG4-ROD, IOI, and MALT lymphoma were the most prevalent specific aetiologies. It is difficult to determine the precise etiology of the symptoms based only on clinical presentation because of the diverse spectrum of potential aetiologies. Therefore, in the majority of patients presenting with bilateral lacrimal gland lesions, a biopsy should be done to accurately determine the etiology.

## Investigations

The analysis of this case series showed that certain clinical characteristics may assist in distinguishing between various etiologies. For example, IOI was more common in younger patients, lid swelling was more common in IgG4-ROD individuals, and palpable masses were more common in MALT lymphoma patients. These findings differ from those reported in earlier studies [1, 2]. In comparison to the findings of Huang et al., systemic symptoms were far more prevalent in individuals with IgG4-ROD in the present study, and many of these patients had multiple organ involvement [2]. A possible explanation for this inconsistency is that the present study did not classify IgG4-ROD as inflammatory.

Fifteen patients showed signs of unilateral symptoms. Imaging tests identified bilateral lacrimal gland tumors in these individuals, despite their diagnoses of MALT lymphoma (n = 5), IgG4-ROD (n = 4), RLH (n = 3), dacryops (n = 1), amyloidosis (n = 1), and mantle cell lymphoma (n = 1). Therefore, imaging studies are useful for both clinical diagnosis and therapy, and they are crucial in the diagnosis of individuals with bilateral lacrimal gland abnormalities. For example, inflammatory conditions were more likely to manifest with diffuse enlargement of the lacrimal gland with maintenance of the shape, benign tumors were likely to appear as scalloped masses, and malignant tumors were likely to be associated with bone destruction. Additionally, involvement of the extra-lacrimal gland may also be observed, which is particularly helpful in the identification of IgG4-ROD individuals who frequently show swelling of the extraocular muscles and sensory nerves [3]. Imaging scans revealed that 21 individuals had the presence of structures beyond the lacrimal glands, and among them, 13 patients were diagnosed with IgG4-ROD. Therefore, it is suggested that a thorough medical history and extensive examinations should be given priority in the initial diagnostic evaluation.

## Pathological diagnosis

Bilateral lacrimal gland lesions have been reported in several earlier studies. IgG4-ROD, IOID, lymphoma, lacrimal gland prolapse, and sarcoidosis were found to be the most prevalent specific diagnoses in a cohort of 115 patients in a case series by Huang et al. [2]. According to Tang et al., among 97 patients, the most prevalent specific diagnoses were lymphoma, lymphoid hyperplasia, sarcoidosis, IOID, and lacrimal gland prolapse [1]. Other publications on bilateral lacrimal gland lesions are mostly case reports showing associations with Kimura's disease [4–6], lacrimal gland agenesis [7, 8], diffuse large B-cell lymphoma [9], COVID-19 dacryoadenitis [10], tuberculous dacryoadenitis [11], amyloidosis [12], Rosai-Dorfman disease [1, 13], Still's disease [14], TED [15], acute lymphocytic leukemia [16], sickle cell disease [17], and myeloproliferative diseases [18].

In the present series, IgG4-ROD was the most common etiology, followed by IOI, MALT lymphoma, RLH, and mantle cell lymphoma, differing from the findings of Huang et al. [2] and Tang et al. [1]. Further analysis revealed that the incidence of IgG4-ROD and lymphoma was higher than their results, while the incidence of IOI was higher than Huang et al. and slightly lower than Tang et al. The reasons for the discrepancies may be attributed to the use of varying thresholds during the biopsy, epidemiological differences between different ethnicities, or the comprehensive testing of serum IgG4. Additionally, this investigation did not identify any correlation between bilateral lacrimal gland lesions and sarcoidosis, which deviates from the earlier research [1, 2]. Sarcoidosis may be seen more commonly in African-American patients [19]. Furthermore, the present results differed from those of Tao et al. [20] and Gündüz et al. [21]. Tao et al. found that pleomorphic adenoma, adenoid cystic carcinoma, adenocarcinoma, inflammatory pseudotumor, and dermoid cyst were the most common specific diagnoses in 91 cases of lacrimal gland space-occupying lesions. Chronic dacryadenitis, lymphoid tumor, pleomorphic adenoma, adenoid cystic carcinoma, and pleomorphic adenocarcinoma were the most common specific causes in 31 cases of unilateral lacrimal gland lesions by Gündüz et al.. The incidence of inflammation in unilateral lacrimal gland lesions was higher than our series, whereas lymphoid tumors were lower than our results. This difference may be due to the different epidemiology of unilateral and bilateral lacrimal gland lesions and different ethnicities.

## Outcomes

Overall, surgery was the primary method of treatment, followed by surgery in conjunction with corticosteroids. In the present dataset, most patients with IgG4-ROD received treatment with immunosuppressants and corticosteroids in addition to surgery alone. About 5 of these patients showed signs of disease progression; the others had either stable disease, complete remission of the ailment, or significant improvement. The proportions of patients who experienced complete resolution or stable disease were lower than those reported by Huang et al., but the number of patients demonstrating significant improvement was higher. These outcome proportions are consistent with those found in prior studies [1, 2]. Furthermore, it was discovered that patients with IOI had a shorter recurrence-free period than those with IgG4-ROD and MALT lymphoma, with IOI recurrence occurring at 15 months following surgery.

## Management

The observation that IgG4-ROD was the most common specific etiology associated with bilateral lacrimal gland lesions in the present population, may suggest that serum IgG4 levels should be analyzed in all patients presenting with bilateral lacrimal gland lesions. However, it

should be noted that some patients with biopsy-proven IgG4-ROD have normal levels of serum IgG4 [22]. Therefore, this study suggests carrying out a biopsy with IgG4 staining for all patients, regardless of whether their serum IgG4 levels are within the normal range. Biopsy is highly recommended for individuals who have bilateral lacrimal gland lesions, particularly when they demonstrate systemic symptoms. However, specific clinical criteria can help make a diagnosis.

This study has several limitations. First, many etiologies of bilateral lacrimal gland lesions are still extremely rare, with only a few patients in our series, which limits our ability to summarize the clinical characteristics of rare diseases. Secondly, these data may not represent the majority of Chinese lacrimal gland patients, and further research is needed for larger and more extensive data. Despite these limitations, this is the first to evaluate the clinicopathological features and outcomes of bilateral lacrimal gland lesions in an Asian population.

## Conclusion

Overall, bilateral lacrimal gland lesions are associated with a diverse range of causes, most of which are systemic disorders. The study's findings revealed that IgG4-ROD was the most prevalent specific etiology. The present study has identified some clinical criteria that can help in distinguishing between these lesions. Nevertheless, as a result of the complex origins of bilateral lacrimal gland lesions, it is extremely hard to ascertain the precise cause of the problem just based on the patient's clinical characteristics. Therefore, to determine the precise cause, it is crucial to utilize biopsy as a significant tool in patients who show bilateral lacrimal gland abnormalities.

## Supporting information

**S1 Raw data.**
(XLSX)

## Acknowledgments

The author acknowledge all the clinicians who provided us with assistance during the data collection.

## Author Contributions

**Conceptualization:** Lvfu He, Weimin He.

**Data curation:** Lvfu He.

**Formal analysis:** Lvfu He.

**Investigation:** Lvfu He, Weimin He.

**Methodology:** Lvfu He, Weimin He.

**Resources:** Lvfu He, Weimin He.

**Software:** Lvfu He.

**Supervision:** Weimin He.

**Validation:** Lvfu He.

**Visualization:** Lvfu He.

**Writing – original draft:** Lvfu He.

**Writing – review & editing:** Lvfu He.

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
