## [Decision Letter · Decision Letter 0]

20 Feb 2024

PONE-D-23-44008Clinicopathologic features and outcomes of bilateral lacrimal gland lesionsPLOS ONE

Dear Dr. He,

Thank you for submitting your manuscript to PLOS ONE. After careful consideration, we feel that it has merit but does not fully meet PLOS ONE’s publication criteria as it currently stands. Therefore, we invite you to submit a revised version of the manuscript that addresses the points raised during the review process.

**ACADEMIC EDITOR: **In my opinion and based on the reviewers` comments the manuscript needs revision. There are some points within the manuscript which must be corrected. The lack of clear Inclusion/Exclusion criteria with inclusion of previously treated patients are major drawbacks on the validity and reliability of the results and outcomes. The paper needs extensive English editing. Please find the attached files with reviewers` comments and carefully addressed all the suggestions.

We look forward to receiving your revised manuscript.

Kind regards,

Karolina Goździewska-Harłajczuk

Academic Editor

PLOS ONE

Journal Requirements:

2. In this instance it seems there may be acceptable restrictions in place that prevent the public sharing of your minimal data. However, in line with our goal of ensuring long-term data availability to all interested researchers, PLOS’ Data Policy states that authors cannot be the sole named individuals responsible for ensuring data access (http://journals.plos.org/plosone/s/data-availability#loc-acceptable-data-sharing-methods).

3. We note that Figure [1] in your submission contain copyrighted images. All PLOS content is published under the Creative Commons Attribution License (CC BY 4.0), which means that the manuscript, images, and Supporting Information files will be freely available online, and any third party is permitted to access, download, copy, distribute, and use these materials in any way, even commercially, with proper attribution. For more information, see our copyright guidelines: http://journals.plos.org/plosone/s/licenses-and-copyright.

a. You may seek permission from the original copyright holder of Figure [1] to publish the content specifically under the CC BY 4.0 license. 

4. Please include your tables as part of your main manuscript and remove the individual files. Please note that supplementary tables (should remain/ be uploaded) as separate "supporting information" files.

Additional Editor Comments:

In my opinion and based on the reviewers` comments the manuscript needs revision. There are some points within the manuscript which must be corrected. The lack of clear Inclusion/Exclusion criteria with inclusion of previously treated patients are major drawbacks on the validity and reliability of the results and outcomes. The paper needs extensive English editing. Please find the attached files with reviewers` comments and carefully addressed all the suggestions.

Reviewers' comments:

Reviewer's Responses to Questions

**Comments to the Author**

1. Is the manuscript technically sound, and do the data support the conclusions?

Reviewer #1: Yes

Reviewer #2: Partly

Reviewer #3: Yes

2. Has the statistical analysis been performed appropriately and rigorously? 

Reviewer #1: Yes

Reviewer #2: No

Reviewer #3: Yes

3. Have the authors made all data underlying the findings in their manuscript fully available?

Reviewer #1: Yes

Reviewer #2: Yes

Reviewer #3: No

4. Is the manuscript presented in an intelligible fashion and written in standard English?

Reviewer #1: No

Reviewer #2: Yes

Reviewer #3: Yes

5. Review Comments to the Author

Reviewer #1: This study duplicates to a certain extent two other studies looking at bilateral lacrimal gland involvement by various diseases (both cited by these authors), but adds to that knowledge base by presenting a series from China.

There are quite a number of things that could be improved to make the paper more presentable and understandable. In general, the English is not good, and the paper needs extensive editing. Services are readily available online, and it is beyond the remit of the reviewers and editors of the journal to make those changes.

Abstract - Results - "followed by dacryoadenitis" should have "non-specific" added before "dacryoadenitis"

- "inflammatory pseudotumor". This is not a diagnosis now supported by most specialists working in this field. Idiopathic orbital inflammation is a better term. How is this different to non-specific dacryoadenitis? What criteria were used to make this diagnosis and how was it separated from non-specific dacryoadenitis?

Results section of the paper:

- Demographics - what does "hormone therapy" mean?. Please specify what this is.

- Clinical presentation - what is "hypostasis"?

- Laboratory investigations - "the patient with MALT ....", and "The patient with elevated ...." Do you mean just one patient each time, or more than one?

- Radiological investigation - what does "long or equal on both T1 and T2" mean? Please clarify.

- Histopathology classification - what is the difference between dacryoadenitis (presumably non-specific) and inflammatory pseudotumor?

- Treatment - most common was surgery combined with corticosteroids. What exactly was entailed with the surgical part of this? Was it just incisional biopsy, or intentional debulking surgery? Incisional biopsy alone cannot be regarded as "treatment", but is rather an investigation.

Discussion -

- page 13 - the list of "mainly case reports" is misleading. There is a much larger literature than "mainly case reports" for diseases such as sarcoidosis, Erdheim-Chester disease, Sjögren's syndrome.

- page 14 - the paragraph starting "in addition, biopsy should be strongly ..." Why do you then suggest biopsy is NOT recommended in the "effectively treated"? Are some patients treated speculatively without a biopsy?

Images - the histopathological images are of too low power and size to interpret, and add very little to the paper. I would delete them.

Tables - Table 2, 3 and 4 (two last columns) suggest that percentages be added in brackets after each raw number. The raw numbers are relatively meaningless without the percentages.

Reviewer #2: The authors stated in the Results section:

"In addition, 36 patients(27.9%) previously performed relevant treatment,including hormone therapy(n=16),excision of orbital mass (n=8) and submandibular gland(n=4) and eyelid mass (n=2)and conjunctival mass (n=1), fixation of lacrimal gland prolapse(n=2), double eyelid operation (n=1), radiotherapy (n=1) and chemotherapy(n=1)(Table 1)."

This is confusing since 16 patients had hormone therapy presumably for TED, however, the final results of the study didn't show TED cases. Also the authors didn't have well defined Inclusion/Exclusion criteria apart from "bilaterality".

The Bilaterality in some of the cases was identified only by Radiology, which was not clarified in the methodology.

The authors included patients with H/O hormone therapy, radiotherapy, and chemotherapy as well as previous excision of orbital mass.

All the above has negative impact on the validity and reliability of the results and outcome.

It might be wise to exclude those patients with previous therapy and to have the analysis repeated for the cases with primary bilateral LG lesions with no prior medical/surgical interference related to the original pathologies.

The manuscript can be considered after that.

Reviewer #3: 1. Check reference no. 1. The format is not consistent with journal format.

2. Tables 1-5 were not included in the manuscript. Only the figures are available.

I can't relate to the discussion and conclusion.

6. PLOS authors have the option to publish the peer review history of their article (what does this mean?). If published, this will include your full peer review and any attached files.

Reviewer #1: No

Reviewer #2: **Yes: **Hind M Alkatan

Reviewer #3: **Yes: **Shatriah Ismail

---

## [Decision Letter · Decision Letter 1]

17 May 2024

PONE-D-23-44008R1Clinicopathologic features and outcomes of bilateral lacrimal gland lesionsPLOS ONE

Dear Dr. He,

Thank you for submitting your manuscript to PLOS ONE. After careful consideration, we feel that it has merit but does not fully meet PLOS ONE’s publication criteria as it currently stands. Therefore, we invite you to submit a revised version of the manuscript that addresses the points raised during the review process.

**ACADEMIC EDITOR:**

The manuscript was partly improved, however based on Reviewers` suggestions there are still some points which need a general revision.

We look forward to receiving your revised manuscript.

Kind regards,

Karolina Goździewska-Harłajczuk

Academic Editor

PLOS ONE

Additional Editor Comments:

The manuscript was partly improved, however based on Reviewers` suggestions there are still some points which need a general revision.

Reviewers' comments:

Reviewer's Responses to Questions

**Comments to the Author**

1. If the authors have adequately addressed your comments raised in a previous round of review and you feel that this manuscript is now acceptable for publication, you may indicate that here to bypass the “Comments to the Author” section, enter your conflict of interest statement in the “Confidential to Editor” section, and submit your "Accept" recommendation.

Reviewer #2: All comments have been addressed

Reviewer #4: (No Response)

Reviewer #5: (No Response)

2. Is the manuscript technically sound, and do the data support the conclusions?

Reviewer #2: Yes

Reviewer #4: Partly

Reviewer #5: Partly

3. Has the statistical analysis been performed appropriately and rigorously? 

Reviewer #2: Yes

Reviewer #4: I Don't Know

Reviewer #5: N/A

4. Have the authors made all data underlying the findings in their manuscript fully available?

Reviewer #2: Yes

Reviewer #4: Yes

Reviewer #5: Yes

5. Is the manuscript presented in an intelligible fashion and written in standard English?

PLOS ONE does not copyedit accepted manuscripts, so the language in submitted articles must be clear, correct, and unambiguous. Any typographical or grammatical errors should be corrected at revision, so please note any specific errors here

Reviewer #2: Yes

Reviewer #4: No

Reviewer #5: No

6. Review Comments to the Author

Reviewer #2: (No Response)

Reviewer #4: This is just the review of a hospital with 129 patients only.

There is no new information in this article

Reviewer #5: Although, according to the authors, no previous study has been published on the clinicopathological characteristics of bilateral lacrimal gland lesions biopsied in Asian populations, this paper has certain shortcomings that are markedly important. The quality of written English language text is at times suboptimal. The authors should specify some details about the referral West China Hospital of Sichuan University and the characteristics of the population it serves.

Furthermore, to enrich the content of the discussion, the authors could have compared the pattern of "bilateral" LG involvement with "unilateral" LG involvement data of their region.

The discussion implies that there are two main articles about the characteristics of bilateral LG involvement (by Huang et al in Australia and Tang et al in the US). A more detailed comparison of the findings of the present study with the two other similar case series would have been more informative.

Finally, as the authors have mentioned, the differences in the design of these studies prevent drawing conclusions from such comparison as, for instance, the IgG4 level testing may have been inconsistent among these studies. In this manuscript, only 31 of the 113 cases were tested for serum igG4 levels.

7. PLOS authors have the option to publish the peer review history of their article (what does this mean?). If published, this will include your full peer review and any attached files.

Reviewer #2: **Yes: **Prof. Hind M Alkatan

Reviewer #4: No

Reviewer #5: No

---

## [Author Response · Author response to Decision Letter 1]

25 May 2024

I am very grateful to the reviewers and the editorial comments for their valuable feedback. I have made detailed revisions according to your suggestions. Thank you very much！

---

## [Decision Letter · Decision Letter 2]

2 Jun 2024

PONE-D-23-44008R2Clinicopathologic features and outcomes of bilateral lacrimal gland lesionsPLOS ONE

Dear Dr. He,

Thank you for submitting your manuscript to PLOS ONE. After careful consideration, we feel that it has merit but does not fully meet PLOS ONE’s publication criteria as it currently stands. Therefore, we invite you to submit a revised version of the manuscript that addresses the points raised during the review process.

**ACADEMIC EDITOR: ** The manuscript was improved by the Authors, however the paper still needs minor revision. Please check the comments of the Reviewer and make the necessary correction.

We look forward to receiving your revised manuscript.

Kind regards,

Karolina Goździewska-Harłajczuk

Academic Editor

PLOS ONE

Journal Requirements:

Additional Editor Comments:

The manuscript was improved by the Authors, however the paper still needs minor revision. Please check the comments of the Reviewer and make the necessary correction.

Reviewers' comments:

Reviewer's Responses to Questions

**Comments to the Author**

1. If the authors have adequately addressed your comments raised in a previous round of review and you feel that this manuscript is now acceptable for publication, you may indicate that here to bypass the “Comments to the Author” section, enter your conflict of interest statement in the “Confidential to Editor” section, and submit your "Accept" recommendation.

Reviewer #1: All comments have been addressed

Reviewer #5: (No Response)

2. Is the manuscript technically sound, and do the data support the conclusions?

Reviewer #1: Yes

Reviewer #5: Yes

3. Has the statistical analysis been performed appropriately and rigorously? 

Reviewer #1: Yes

Reviewer #5: Yes

4. Have the authors made all data underlying the findings in their manuscript fully available?

Reviewer #1: Yes

Reviewer #5: Yes

5. Is the manuscript presented in an intelligible fashion and written in standard English?

Reviewer #1: Yes

Reviewer #5: Yes

6. Review Comments to the Author

Reviewer #1: The reviewer's comments have been addressed adequately and the manuscript is now suitable for publication

Reviewer #5: The revised paper is relatively improved and it has merit because of scarcity of similar studies conducted in its corresponding region (west China). However, the reader cannot find out how the "bilateral" lacrimal gland disorders compare to "unilateral" lacrimal gland involvement. So, I recommend the authors to present a very clear comparison between the two.

7. PLOS authors have the option to publish the peer review history of their article (what does this mean?). If published, this will include your full peer review and any attached files.

Reviewer #1: No

Reviewer #5: No

---

## [Author Response · Author response to Decision Letter 2]

3 Jun 2024

Journal Requirements:

Answer:

My reference list conforms to the reference format of plos one. All references were in correct format and did not cite a retracted article.

Reviewer #5: The revised paper is relatively improved and it has merit because of scarcity of similar studies conducted in its corresponding region (west China). However, the reader cannot find out how the "bilateral" lacrimal gland disorders compare to "unilateral" lacrimal gland involvement. So, I recommend the authors to present a very clear comparison between the two. 

Answer:

In our area, I searched papers over the last 30 years and did not find a paper on unilateral lacrimal gland lesions. In other areas, I searched papers for the past 50 years and they were almost case reports. However, I just found a paper on unilateral lacrimal gland lesions. Therefore, in the discussion, I added a comparison between bilateral lacrimal gland lesions and unilateral lacrimal gland lesions. Correspondingly,in the references, I have added a reference(Gündüz K , Shields CL, Günalp I, Shields JA. Magnetic resonance imaging of unilateral lacrimal gland lesions. Graefes Arch Clin Exp Ophthalmol. 2003 Nov;241 (11): 907-913. doi: 10.1007/ s00417-003-0748-z.)

---

## [Editor Report · Decision Letter 3]

5 Jun 2024

Clinicopathologic features and outcomes of bilateral lacrimal gland lesions

PONE-D-23-44008R3

Dear Dr. He,

We’re pleased to inform you that your manuscript has been judged scientifically suitable for publication and will be formally accepted for publication once it meets all outstanding technical requirements.

Kind regards,

Karolina Goździewska-Harłajczuk

Academic Editor

PLOS ONE

Additional Editor Comments (optional):

All the corrections were included within the revised version of the manuscript.
---

## [Editor Report · Acceptance letter]

24 Jun 2024

PONE-D-23-44008R3 

PLOS ONE

Dear Dr. He, 

I'm pleased to inform you that your manuscript has been deemed suitable for publication in PLOS ONE. Congratulations! Your manuscript is now being handed over to our production team.

Kind regards, 

on behalf of

Dr. Karolina Goździewska-Harłajczuk 

Academic Editor

PLOS ONE